# Green Synthesis of Uncoated and Olive Leaf Extract-Coated Silver Nanoparticles: Sunlight Photocatalytic, Antiparasitic, and Antifungal Activities

**DOI:** 10.3390/ijms25063082

**Published:** 2024-03-07

**Authors:** Nasser F. Alotaibi, Laila S. ALqarni, Samia Q. Alghamdi, Sameera N. Al-Ghamdi, Touseef Amna, Soad S. Alzahrani, Shaima M. Moustafa, Tamer H. Hasanin, Amr Mohammad Nassar

**Affiliations:** 1Department of Chemistry, College of Science, Jouf University, Sakaka 42421, Saudi Arabia; nfalotaibi@ju.edu.sa (N.F.A.); thhasanin@ju.edu.sa (T.H.H.); 2Chemistry Department, College of Science, Imam Mohammad Ibn Saud Islamic University (IMSIU), Riyadh 11623, Saudi Arabia; lassalqarni@imamu.edu.sa (L.S.A.); soalzahrani@imamu.edu.sa (S.S.A.); 3Department of Biology, Faculty of Science, Al-Baha University, Al-Baha 65799, Saudi Arabia; sqassim@bu.edu.sa (S.Q.A.); ahaider@bu.edu.sa (T.A.); 4Chemistry Department, Faculty of Science, Al-Baha University, Al-Baha 65799, Saudi Arabia; snasser@bu.edu.sa; 5Department of Biology, College of Science, Jouf University, Sakaka 42421, Saudi Arabia; shymaa.nabil@ju.edu.sa

**Keywords:** Ag-NPs, green synthesis, antiparasitic, antimicrobial, photocatalyst

## Abstract

The circular economy, which attempts to decrease agricultural waste while also improving sustainable development through the production of sustainable products from waste and by-products, is currently one of the main objectives of environmental research. Taking this view, this study used a green approach to synthesize two forms of silver nanoparticles: coated silver nanoparticles with olive leaf extract (Ag-olive) and uncoated pure silver nanoparticles (Ag-pure), which were produced by the calcination of Ag-olive at 550 °C. The extract and the fabricated nanoparticles were characterized by a variety of physicochemical techniques, including high-performance liquid chromatography (HPLC), thermal gravimetric analysis (TGA), X-ray diffraction (XRD), scanning electron microscopy (SEM), and transmission electron microscopy (TEM). Adult ticks (*Hyalomma dromedarii*) (Acari: Ixodidae) were used in this study to evaluate the antiparasitic activity of synthesized nanoparticles and extract. Furthermore, the antifungal activity was evaluated against *Aspergillus aculeatus* strain N (MW958085), *Fuserium oxysporum* (MT550034), and *Alternaria tenuissiuma* (MT550036). In both antiparasitic and antifungal tests, the as-synthesized Ag-olive showed higher inhibition activity than Ag-pure and olive leaf extract. The findings of this research suggest that Ag-olive may be a powerful and eco-friendly antiparasitic and antifungal agent. Ag-pure was also evaluated as a photocatalyst under sunlight for the detoxification of Eri-chrome-black T (EBT), methylene blue (MB), methyl orange (MO), and rhodamine B (RhB).

## 1. Introduction

Green nanotechnology involves using natural bioactive agents, including microbes, plant materials, and biowastes, in the biosynthesis of nanomaterials and for enhancing sustainability by employing nano-products. As a result, it is regarded as an essential part of clean technologies that recover the environment and transform waste bioactive products into more valuable and environmentally friendly green nanomaterials [1,2]. The properties of nanoparticles include being highly mobile when in a free state [3,4]. Also, their surface areas are enormous and specific and exhibit quantum effects [5,6]. Nanoparticles are essential considering their crucial role in modern medicine as they are used in clinical areas [7], including acting as contrast agents during imaging and delivering genes and drugs into tumors [8]. They have important applications in boosting chemical reactions by increasing catalysis [9]. Therefore, nanoparticles assist in reducing the catalytic materials to necessitate the production of desired results that are economically viable and reduce pollutants [10]. They are significantly used in automotive catalytic converters and petroleum refining [11]. The sustainability of nanoparticles is based on the fact that they entail using renewable materials in production, with little influence on the environment [12]. They offer economically efficient and environmentally acceptable solutions for the production of energy and the treatment of waste.

Green nanotechnology involves multiple biological activities [13]. Due to the variety of bioactivities that plant extracts, which include polyphenols, terpenoids, sugars, phenolic acids, proteins, and alkaloids, contain, biological activity involves using these compounds in green synthesis [14,15]. The nanoparticle formation process entails mixing the metal–salt solution with the plant extract sample to reduce the metallic ions and stabilize them [16]. The salt solution is then reduced biochemically, resulting in a color change of the reaction mixture to form the nanoparticles.

The silver nanoparticles Ag-NPs play an important part in desirable inorganic substances. It is crucial to note that Ag-NPs are made in a variety of sizes and will have varying molecular reactivity characteristics based on their surface-area-to-mass ratio [17]. There are numerous ways (i.e., biologically, physically, and chemically) to synthesize nanoparticles (Ag-NPs) [18]. They have a wide range of applications in areas like photography, catalysis, biosensors, bimolecular detection, diagnostics, and especially antibacterial properties. They are also environmentally friendly [19]. Ag-NP has also demonstrated notable promise for enzyme inhibition, antibacterial, antifungal, antioxidant, anti-inflammatory, and anticancer effects. Reactive oxygen species (ROS) produced by Ag-NP have been demonstrated to be harmful to a range of microbial agents and microalgae [20]. In the biogenic synthesis of Ag-NP, several green sources have been used to prepare Ag-NPs [21].

Cultivated since antiquity, olive (*Olea europaea*) trees are evergreen plants with numerous applications in traditional medicine. Oliver leaves are among the world’s most important industrial crops [22]. Olive leaves are well known for their numerous excellent pharmacological effects, or favorable qualities, because of their high concentration of bioactive compounds. According to reports, olive leaves are abundant in a range of bioactive substances [23]. Each of these chemicals possesses exceptional biological characteristics, such as antiviral, antibacterial, anti-inflammatory, hypoglycemic, and antioxidant actions [24].

*Hyalomma dromedarii*, the major species of tick found in Saudi Arabia, affects cattle, camels, and goats and diminishes their production of meat and milk. It also causes weight loss in these animals. Parasite control typically involves the use of chemical acaricides. Using these drugs, however, presents a variety of risks, such as the development of chemical resistance in the tick population, contaminated food and the environment, and adverse bystander effects on animals that are not the intended targets, such as humans. Concerns regarding the safety of these chemical acaricides have led to a search for sustainable and alternative forms of biological tick control that employ plant-derived materials [25,26,27].

Silver nitrate reacts with an extract of olive leaf waste to produce Ag-NPs, which have shown a range of biological, antioxidant, and antibacterial properties against multi-drug-resistant (MDR) bacteria and fungi, as well as anticancer activity against cancer cell lines [28,29]. Ag-NPs were produced using a green synthetic method that used olive leaf extract. It has been researched and optimized to determine which factors, including temperature, exposure duration to the extract, pH, silver nitrate concentration, and the proportion of plant sample to extraction solvent, affected the synthesis of nanoparticles. Investigated were the cytotoxic effects of synthetic Ag-NPs and olive leaf extract containing Ag-NPs on the MCF-7 human breast cancer cell line [30].

The present research focused on the green fabrication of Ag-olive employing olive leaf aqueous extract, which is made up of Ag-NPs with phyto-olive leaf compounds. Ag-pure also emerged following the elimination of biochemicals from Ag-olive. The characterization of the produced NPs, as well as investigations into their antiparasitic, antifungal, and photocatalytic properties, were studied.

## 2. Results and Discussion

The goal of this study was to determine how agricultural waste can be recycled and used as renewable bioresources to produce valuable products that can be employed in environmental and biomedical fields. Ag-olive, a mixture of Ag-NPs and olive leaf extract biochemicals, is synthesized using the aqueous extract of wasted olive leaves. The heating of Ag-olive resulted in the formation of Ag-pure after the removal of bio-organic molecules. The synthesized materials were used in water treatment, antimicrobial, and antiparasitic applications (Figure 1).

### 2.1. Characterization

HPLC was used in this study to investigate the constituents in the aqueous extract of olive leaves. These components were identified using three wavelengths: 280, 320, and 360 nm (Figure 1). The sample was prepared using the procedure described in Section 2.3 and then passed through a 0.22 µm filter for filtration. As shown, HPLC results indicate that protocatechuic acid, p-hydroxybenzoic acid, catechin, caffeic acid, cinnamic acid, rutin, ferulic acid, oleeuropein, luteolin, quercetin, and cinnamic acid were the main components found in the extract. The levels of these components match those identified in the literature [31,32,33].

Powder X-ray diffraction (XRD) was used to analyze the purity and crystallinity of the Ag-olive and Ag-pure, as shown in Figure 2. The peaks with an amorphous phase structure can be seen in the XRD patterns of Ag-olive, which are attributed to the presence of organic phytochemicals in the coating of Ag-NPs [34]. In contrast, the great purity and crystallinity of single-phase Ag-NPs are shown by a strong peak in the XRD patterns of Ag-pure. The peaks observed at 2θ = (111), (200), (220), (311), and (222) crystal planes may easily be identified as the peak positions appearing at 2 = 37.20, 43.20, 62.87, 75.20, and 79.38, respectively, indicating the formation of metallic silver nanoparticles [35]. The face-centered cubic (FCC) crystalline structure of silver, which is the same as that of the reference Ag-NPs (JCPDS, No. 04-0835), can be accurately indexed to each of these diffraction peaks [36]. The samples are single-phase, according to the XRD pattern of Ag-pure, and no other impurities were found. The Scherrer equation, Equation (1), was used to calculate the crystal size [37].
(1)$$D=\frac{0.9\;\cdot \;\lambda \;}{\beta \;\cdot \;Cos\theta \;},$$
where *θ* is the XRD angle, *β* is the half-maximum width, and *λ* = 1.5418 Å is the wavelength of the X-ray (for Cu Kα1). The Ag-pure crystal’s size was calculated to be 13.85 nm.

Ag-olive and Ag-pure were evaluated for purity and thermal stability using TGA. The elimination of the bioorganic molecules (56.50% of the initial sample weight) shows the weight loss observed in the TGA curve of Ag-olive (Figure 3A) within the temperature range of 25–530 °C. Ag-NPs pure weight accounted for 43.50% of the initial sample weight. The TGA for Ag-pure (Figure 3B) shows no thermal degradation and a lack of organic contaminants, indicating the high level of purity of the Ag-NPs.

The morphology of Ag-olive and Ag-pure was investigated using SEM imaging (xT microscope Server) (Figure 4). Morphological studies show the asymmetrical structure of Ag-olive, Figure 4a,b due to the presence of organic compounds derived from olive phytochemicals. A regular, semi-spherical, and polydispersed structure with a diameter ranging from 48 to 95 nm was visible in the Ag-pure image (Figure 4d,e). The presence of C, O, N, P, and S in the organic residues from the extraction of olive leaves is visible in the EDX spectrum of Ag-olive (Figure 4c), while these elements are absent in the spectrum of Ag-pure (Figure 4f), demonstrating the purity of Ag-pure after calcination.

Figure 5A,B shows the obtained TEM images of Ag-olive and Ag-pure, respectively. The TEM photos revealed that Ag nanoparticles had spherical and semi-spherical shapes, were evenly dispersed, had a smooth surface, and were of uniform size. A few tiny particles may aggregate into secondary particles due to their tiny sizes and high surface energy.

It is possible to explain the proposed mechanism of Ag-NP synthesis using wasted olive leaf extract (Figure 6) by pointing out that phytochemical-induced silver ions reduction reactions result in cluster formation, which, in turn, drives nanoparticle growth [38].

### 2.2. Antifungal Activity

Table 1 and Table 2 present the antifungal effects of Ag-olive, Ag-pure, and olive leaf extract, whereas Figure 7 demonstrates these results. The findings were as follows: Ag-olive > Ag-pure > extract from olive leaves. Ag-olive’s potent antifungal action might be attributed to many mechanisms, including the attachment of nanoparticles to fungal surfaces, disruption of cell membranes, alteration of the respiration chain and other permeability-dependent processes, and ultimately, the destruction of vital proteins in microorganisms [38]. Figure 8 displays the suggested mechanism of bioactivity of Ag-NPs. Ag-NPs form complexes with fungal nucleic acids through interactions and changes to cell permeability, allowing proteins, carbohydrates, and nuclear material to all pass through the damaged membrane. It can impede the release of adenosine triphosphate (ATP) and instead produce reactive oxygen species (ROS) [39], which play a key role in a number of DNA functions, such as issues with DNA replication and cell development. By altering the DNA and disrupting the cell membrane, the manufacturing of proteins may be effectively stopped [40].

### 2.3. Antiparasitic Activity

In this study, we assessed the Ag-NPs activity against the active stage (on the host) of *H. dromedarii* adults, as mentioned in the methods section. Ag-NPs were given in varying amounts to the ticks. To determine the minimal acaricidal concentration, three replicate tests were carried out for each given concentration. Every day, the ticks were monitored, and the number of living ticks was noted. The total duration of the experiment persisted from 0 to 7 days. Figure 9 illustrates the dose-dependent observation of the mortality percentage. After seven days of incubation, no deaths were seen in the untreated control tick population.

The antiparasitic studies exposed that the maximum mortality % was detected at the highest concentration (0.06 mg/mL) after five days of incubation period. Using of plant material for synthesis of nanoparticles is speedy, economical, eco-friendly and a facile one-way procedure for the biosynthesis [21]. Amongst the numerous acknowledged synthesis procedures, the plant-mediated nanoparticle preparation is ideal, as it is cost-effective and environmentally responsive, as well as safe for human therapeutic usage [22]. Nonetheless, in recent times, numerous studies have been carried out to observe Ag toxicity against a number of pests and ticks. However, to the best of our information, there is no report about the acaricidal activity against the adult camel ticks of *H. dromedarii*. In the present study waste, olive leaves of Al-Baha were utilized for the synthesis of Ag-NPs, and these nanoparticles were tested for acaricidal activity. Although a number of studies have been conducted on nanoparticles, the understanding of comprehensive mechanisms of action of NPs against aforementioned insects and mites is inadequately understood. Various modes of action have been expected through which the nano-pesticides act on ticks, for instance, upsetting the metabolism, changing the structure of lipids or proteins, and producing oxidative stress in ticks. Moreover, these NPs tend to bind to biomolecules such as amino acids, proteins, and nucleic acids, shrinking the membrane permeability and eventually causing death [12]. 

Recently, a report has been published about the cytotoxicity and genotoxicity of Ag NPs allocated to their surface properties related to toxicity in organic models [41]. Considering the noteworthy properties of Ag-NPs, our work assesses the dose dependence of Ag NPs action on *H. dromedarii* adult ticks. This work is the first report on the antiparasitic activities of Ag-NPs synthesized from olive leaf extract. Ag-NPs demonstrated the dose-dependent mortality. Further study will be conducted on the exogenous acaricidal activity under in vivo conditions.

It has been proposed that host cell biological processes are accelerated or decelerated by nanoparticles, which causes toxicity [42]. Nanomaterials’ distinct physicochemical surface features make them more appropriate for downstream functionalized applications [43]. Moreover, metal nanoparticles have been shown to have effective antibacterial effects due to their unique characteristics and vast surface areas [44]. ROS (Reactive oxygen species) that have the ability for eliminating harmful microbes because of their high chemical reactivity, may be produced by nanoparticles in particular. As a result, nanoparticles are suggested for parasite destruction (cytotoxic and inhibitory action) because they serve as more effective and less damaging medications [45]. There would be a greater incentive to fabricate eco-friendly nanoparticles that are fewer harmful to the environment than traditional formulations, and additional study would be necessary to find any usable products that might compete with present structures in terms of both cost and efficacy [46]. The tick is affected by nanopesticides in a variety of ways, such as by changing lipids or proteins, inducing oxidative stress, or interfering with tick and microbe metabolisms [47]. The bioactivities of Ag-olive are related to similar works in Table 3.

### 2.4. Photocatalytic Studies

Due to an increased demand for clean water resources, the practice of purifying and treating water has rapidly spread throughout the world [57]. Ag-pure is assessed as a photocatalyst under solar irradiation. A total of 5 ppm solutions of two cationic dyes (MB and RhB) and two anionic dyes (MO and EBT) were employed for the photocatalytic reactions. Three distinct control experiments were conducted to examine the absorption spectra of the studied dyes: catalyst adsorption in the absence of sunlight, photolysis in the absence, and the presence of Ag-pure. The photolysis experiment showed that, in the absence of a catalyst, there was no color variation. In the absence of sunlight, the adsorption experiments with RhB, MB, and EBT showed removal efficiencies of 66%, 55%, and 49%, respectively, after 15 min. This suggests the existence of active sites that enabled the dye to adhere to the Ag-pure surface. MO dye showed no color change in the presence or absence of sunlight.

Under sunlight, the dyes began to decrease in color after 2 min and completely after 5 min. The removal efficiency for each dye was calculated using Equation (2) [58].
(2)$$\mathrm{Degradation}\;(\%)=\frac{A_{0}-A_{t}}{A_{0}}\;\times \;100,$$
where *A*_0_ is the absorbance value of the dye solution before treatment, and *A_t_* is the absorbance value after treatment.

Ag-pure showed very high activity in the photocatalytic degradation of RhB, MB, and EBT, with removal efficiencies of 96%, 95%, and 85%, respectively, after only 5 min at natural pH and room temperature calculated from the absorbance of the chromophore peak of each dye (Figure 10). In light of these studies, Ag-pure has been determined to be a solar semiconductor that efficiently slows down the decomposition of MB, RhB, and EBT when exposed to sunlight (Figure 11).

In the instance of MO, there was no noticeable color change, suggesting that this dye was not degraded. This was further supported by the fact that the absorbance of the spectrophotometer’s chromophore peak remained unshifted.

The photocatalytic results of Ag-pure are compared to other biosynthesized nanoparticles in Table 4.

## *3.* Materials and Methods

### 3.1. Materials

Wasted olive leaves (*Olea europaea* L.) were gathered in a public garden in Al-Baha, Saudi Arabia. Dr. Najla A. Al-Shaye identified the plant specimen, which was stored in the Princess Nourah bint Abdulrahman University Herbarium (voucher number 6217 PNUH). Organic dyes, erichrome black T (EBT), methylene blue (MB), methyl orange (MO), silver nitrate, and rhodamine B (RhB), were supplied by Sigma-Aldrich (USA). Every aqueous solution was prepared with double-distilled water. The synthesized materials were evaluated for their antimicrobial activity against the fungal species *Aspergillus aculeatus* strain N (MW958085), *Fuserium oxysporum* (MW830121), and *Alternaria tenuissiuma* (MT550036). These isolates were isolated from water samples. Adult ticks *(Hyalomma dromedarii*) (Acari: Ixodidae) used in the laboratory and field assessments were collected from camels (Camelus dromedarius) not treated with any pesticides from Al-Baha, Saudi Arabia. A total of 300 ticks were kept in a dry Eppendorf tube at room temperature.

### 3.2. Instruments

Using a Thermo Scientific Quattro S system (Waltham, MA, USA), SEM photodetection was performed. X-ray diffraction (XRD) patterns were detected using the XRD-7000 SHIMADZU by means of a copper radiation source (Kyoto, Japan). Thermal gravimetric analysis (TGA) was detected using the TGA-51SHIMADZU at a heating rate of 10 °C/min. A JEOL GEM-1010 transmission electron microscope running at 80 kV was used to take the transmittance electron microscope (TEM) images. The Labomed-Spector 99 UV-Vis double-beam 3200 (Tokyo, Japan) was used to obtain UV-Vis spectra. An Agilent 1100 model HPLC system, fitted with a Phenomenex Aqua C18 (5 μm) 250 mm × 4.6 mm d. column (Torrance, CA, USA) and a diode array detection system G1315A, was utilized to analyze the extract using high-performance liquid chromatography (HPLC) (Markham, Canada). 

### 3.3. Preparation of Aqueous Extract of Olive Leaf

The collected leaves were thoroughly cleaned with distilled water to get rid of any pollutants before being dried in the air. A total of 5 g of dried leaf was combined with 200 mL of distilled water and then refluxed for two hours at 100 °C to prepare the aqueous extract. The mixture was subsequently cooled to room temperature. Whatman No. 1 filter paper was used to filter the solution, and the filtrate was then employed in the following tests [28].

### 3.4. Synthesis of Ag-Olive and Ag-Pure

Green synthesis of Ag-olive was carried out as follows: 100 mL of doubly distilled water was used to dissolve 3 g of silver nitrate, AgNO_3_, before adding it to the olive leaf extract. For 2 h, the mixture was stirred at 70 °C and 1200 rpm. The formation of a dark brown precipitate revealed that the synthesis of Ag-NPs was complete. The solution was then allowed to cool before being centrifuged three times with double-distilled water at 3000 rpm for 15–25 min. To produce Ag-pure, a portion of the precipitate was finally burned in a furnace for 2 h at 550 °C.

### 3.5. Antifungal Experiments

#### 3.5.1. Solid Media

*Fuserium oxysporum* (MT550034), *Alternaria tenuissiuma* (MT550036), and *Aspergillus aculeatus* strain N (MW958085) were identified from agricultural and groundwater samples. After the Potato-Dextrose-Agar medium was prepared and sterilized, the investigated compounds (Ag-olive, Ag-pure, and olive leaf extract) (100 µL) were added to it and the Petri dishes were filled. Ag-olive, Ag-pure, and olive leaf extract were the three investigated compounds whose concentrations that were evaluated: 60, 90, and 120 parts per million. A 0.5 cm PDA disc was positioned in the middle of each plate, and several tested fungi underwent 6–10 days of mycelial development. Overnight, the plates were kept at 26 °C. There were three copies of every treatment. Only PDA media was used for controls. With the exception of *Alternaria tenuissiuma*, which was evaluated after 10 days, the mycelium growth of the studied fungus was assessed in every plate after 6 days. Using Equation (3), the proportion of mycelial growth inhibition was computed [63].
% Inhibition = (C_mg_ − T_mg_)/C_mg_ × 100,(3)

The diameter of the treatment colony is T_mg_, while the diameter of the control colony is C_mg_ (cm).

#### 3.5.2. In liquid Media

After adding 1 mL of each of the studied compounds (olive leaf extract, Ag-olive, and Ag-pure) at 60, 90, and 120 ppm to 50 mL of PD broth in a 100 mL Erlenmeyer flask, the mixture was autoclaved to ensure sterilization. A disc of a fungal growth from one of the fungi under test was used to inoculate each flask. For 6–10 days, the incubation process was carried out in the dark at 26 °C. Each flask’s growth was filtered using a 9-cm filter paper, and it was then twice cleaned with distilled water before being let to dry for 24 h at room temperature. By deducting the weight of the dry filter paper from the total weight, the growth of each fungus was calculated. PD broth was the sole ingredient used in the controls.

### 3.6. Antiparasitic Activity Experiments

In the present study, three different types of samples, such as olive extract, Ag-pure, and Ag-olive, were tested for antiparasitic activity. In brief, all the samples were mixed with 200 µL of blood in 1.5 mL of Eppendorf tubes and vortex for one min on a twister. The 4000 adult ticks were classified into 4 groups on the basis of the treatment (control, Ag-olive, olive extract, and Ag-pure). The antiparasitic effect of aforementioned samples was tested using the various concentrations (0.01, 0.02, 0.03, and 0.06 µg/mL) for 7 days to check out MIC (minimum inhibitory concentration). The number of adult ticks and mortality rate were also quantified. Ticks in the fourth group were fed simply blood; ticks in the other 3 groups were fed blood mixed with Ag-olive, Ag-pure, and natural olive extract. Adult ticks that had fed with blood only were used as control. Every cultured plate was wrapped with foil, with an opening to let air flow through plastic on top and each dish had a two-day diet plan. Ticks fed with different supplements and control groups were cultured at room temperature (25 °C) with appropriate humidity (61–75%). Up to seven days of incubation, daily data on the death rate was kept. Daily observations were conducted using appropriate changes to previously published procedures, and after the mortality rate reached 50%, the adult tick mortality (%) was recorded [53,64].

### 3.7. Data Analysis

The data displays the standard deviation and mean values for the three replicates. Software called SPSS software (Origin Pro2016) was used to do statistical analysis.

### 3.8. Photocatalytic Activity

Ag-pure was investigated as a photocatalyst under solar radiation for the removal of methylene blue (MB), methyl orange (MO), rhodamine B (RhB), and erichrome black T (EBT). A total of 20 mg of Ag-pure was added to 50 mL of 5 ppm of each dye in a 200 mL beaker flask, and the mixture was swirled for two minutes in the dark to achieve equilibrium for the dyes on the catalyst surface. The dye solutions were subsequently exposed to 40 × 10^3^ LUX of solar radiation for 5 min. Next, in order to separate the catalyst, 3 mL of the treated solutions were centrifuged for 2 min and quantified using a spectrophotometer.

## 4. Conclusions

In conclusion, the present green approach demonstrates wasted olive leaves as a renewable and eco-friendly source for the synthesis of silver nanoparticles. The current work is the first report on the assessments of antiparasitic activities to determine the effectiveness of synthesized Ag-NP prepared from olive leaf extract. Ag-olive was found to be an extremely effective antiparasitic agent against adult ticks (*Hyalomma dromedarii*) (Acari: Ixodidae). Ag-olive showed a high level of antifungal activity against *Aspergillus aculeatus* strain N, *Fuserium oxysporum*, and *Alternaria tenuissiuma*. More research is needed to evaluate these exogenous acaricidal and antifungal activity samples in vivo. Ag-pure exhibited rapid and high photocatalytic activity under solar irradiation to remove eriochrome black T, rhodamine B, and methylene blue. Additionally, further study will be done to determine the ideal conditions for Ag-pure to photodegrade methyl orange dye.

## Data Availability

The raw data for this study will be available upon request.

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
