# Peer review of "Green Synthesis of Uncoated and Olive Leaf Extract-Coated Silver Nanoparticles: Sunlight Photocatalytic, Antiparasitic, and Antifungal Activities"

_ijms, 2024, doi:10.3390/ijms25063082_

Round 1
Reviewer 1 Report
Comments and Suggestions for Authors
In this manuscript, the authors reported green synthesis of uncoated and olive leaf extract-coated Ag nanoparticles, and studied their photocatalytic, antiparasitic, and antifungal performance. This work seems to be useful for this field. However, the following problems should be addressed before further consideration of publication:
1. “Green synthesis” can be a better keyword instead of “sustainable”.
2. The abstract needs revision to be concise. The development and novelty of this work can be stated, especially the superiority when compared with other researches.
3. In the Introduction, the development of biosynthesis and biotemplated fabrication can be briefly revised. The recent advances should be added including: 10.1021/acsami.1c16859, 10.1002/sstr.202200356, 10.1002/EXP.20220072.
4. All the figures need to be revised with improved quality, where consistent layout/size are desired to improve the readability. Some inset notes and figures are not clear.
5. Scheme 1 should be revised to show the detailed process and mechanisms of green synthesis as well as various functional applications.
6. The standard peaks of Ag should also be noted in Figure 2 for better comparison.
7. Except for TEM characterization, HRTEM with detailed lattice analysis should be added.
8. Figure 6 was too simple to verify the mechanisms. In the manuscript, the depth could be improved if the authors provided some insights of micro-/nano interactions responsible for green synthesis and applications of the Ag NPs.
9. The conditions of control tests in Figure 7 should be noted clearly. The results in Figure 9 are also confusing, which can be explained in detail.
10. The reference format should be checked due to some errors.
Author Response
Reviewer #1: In this manuscript, the authors reported green synthesis of uncoated and olive leaf extract-coated Ag nanoparticles, and studied their photocatalytic, antiparasitic, and antifungal performance. This work seems to be useful for this field. However, the following problems should be addressed before further consideration of publication: 1. “Green synthesis” can be a better keyword instead of “sustainable”. - First, we would like to thank the reviewer so much for the valuable comments. - Thank you for pointing it out. In response to your comment, the keyword has been changed. 2. The abstract needs revision to be concise. The development and novelty of this work can be stated, especially the superiority when compared with other researches. - Thank you for pointing it out. In response to your comment, the abstract has been updated. 3. In the Introduction, the development of biosynthesis and biotemplated fabrication can be briefly revised. The recent advances should be added including: 10.1021/acsami.1c16859, 10.1002/sstr.202200356, 10.1002/EXP.20220072. - Thank you for pointing it out. In response to your comment, the references section has been modified. 4. All the figures need to be revised with improved quality, where consistent layout/size are desired to improve the readability. Some inset notes and figures are not clear. - Thank you for pointing it out. In response to your comment, all figures have been updated. 5. Scheme 1 should be revised to show the detailed process and mechanisms of green synthesis as well as various functional applications. - Thank you for pointing it out. In response to your comment, scheme 1 has been updated. 6. The standard peaks of Ag should also be noted in Figure 2 for better comparison. - Thank you for pointing it out. In response to your comment, the standard peaks of Ag-NPs have been illustrated in Fig.2. 7. Except for TEM characterization, HRTEM with detailed lattice analysis should be added. -Thank you for pointing it out. Unfortunately, HRTEM is not available at this time. Further research is needed to study the photocatalytic degradation of methyl orange using Ag-pure as a photocatalyst, and your valuable suggestions will be taken care of. 8. Figure 6 was too simple to verify the mechanisms. In the manuscript, the depth could be improved if the authors provided some insights of micro-/nano interactions responsible for green synthesis and applications of the Ag NPs. - Thank you for pointing it out. In response to your comment,Fig.6 has been updated. 9. The conditions of control tests in Figure 7 should be noted clearly. The results in Figure 9 are also confusing, which can be explained in detail. -Thank you for your comment, in Fig 7, PDA is the negative control without any treatment. In response to your comment, the experimental section on antimicrobial activity has been modified. Also, in response to your comment, the results in Fig 9 are explained in detail. 10. The reference format should be checked due to some errors. - Thank you for pointing it out. In response to your comment, the references have been updated.Reviewer 2 Report
Comments and Suggestions for Authors
In my opinion, the manuscript needs to be extensively modified. There are some points that should be elucidated or improved and the study requires a major revision before publication.
- Antiparasitic activity experiments were not described properly. 200 liters is a large amount of blood to test a single sample. “ 400 adult ticks were divided into four groups” What does it mean? - 4 samples of particles were immersed in blood and then … the ticks were put into the blood. – what does it mean “expose” in this experiment – it should be described in detail. The ticks need to sack blood from the host body. Probably the test found how many ticks survive in liquid and did not drow. The experiment should contain also the sample (control) ticks in blood without AgNPs. The key issue is also how antiparasitic properties of AgNPs will be applied, without in vivo experiments with host organisms is not possible to find proper concentrations of AgNPs, which do not harmfully influence animals.
- The photocatalytic part should be also significantly improved. What was the wavelength of the lamp? The mechanism of photodegradation should be proposed. This part needs essential control experiments to prove the photodegradation process – according to the author's “solar degradation” - two different irradiation sources should be selected: from the visible part of spectra and from UV, the experiment without AgNPs to show that the dyes do not degradation in time or because of temperature? Moreover, an experiment carried out in the dark, without any lamp and irradiation is necessary to prove electromagnetic influence on the process. It is essential to prepare also experiment with commercial photocatalysts – P25 (TiO2) to compare the efficiency of the process. Why did methyl orange not decompose in the presence of AgNPs?
- The authors should explain why the AgNPs size from Scherrer calculations is not comparable to SEM analysis.
- Figures 5A and 5B need separate figure captions.
- Figure 2 – the main bands should be labeled in Figure and It should be explained why some major peaks connected to Ag presence are not visible in the diffractometer of Ag-olive.
The experiment part should be rewritten to apply all experimental details – preparation of the samples for SEM and TEM measurements, “the concentration were 60, 90 and 120 ppm” ppm of what? What does it mean “were isolated from Agriculture” – it should be specified.
Comments on the Quality of English Language
In my opinion, the manuscript needs to be extensively modified. There are some points that should be elucidated or improved and the study requires a major revision before publication.
- Antiparasitic activity experiments were not described properly. 200 liters is a large amount of blood to test a single sample. “ 400 adult ticks were divided into four groups” What does it mean? - 4 samples of particles were immersed in blood and then … the ticks were put into the blood. – what does it mean “expose” in this experiment – it should be described in detail. The ticks need to sack blood from the host body. Probably the test found how many ticks survive in liquid and did not drow. The experiment should contain also the sample (control) ticks in blood without AgNPs. The key issue is also how antiparasitic properties of AgNPs will be applied, without in vivo experiments with host organisms is not possible to find proper concentrations of AgNPs, which do not harmfully influence animals.
- The photocatalytic part should be also significantly improved. What was the wavelength of the lamp? The mechanism of photodegradation should be proposed. This part needs essential control experiments to prove the photodegradation process – according to the author's “solar degradation” - two different irradiation sources should be selected: from the visible part of spectra and from UV, the experiment without AgNPs to show that the dyes do not degradation in time or because of temperature? Moreover, an experiment carried out in the dark, without any lamp and irradiation is necessary to prove electromagnetic influence on the process. It is essential to prepare also experiment with commercial photocatalysts – P25 (TiO2) to compare the efficiency of the process. Why did methyl orange not decompose in the presence of AgNPs?
- The authors should explain why the AgNPs size from Scherrer calculations is not comparable to SEM analysis.
- Figures 5A and 5B need separate figure captions.
- Figure 2 – the main bands should be labeled in Figure and It should be explained why some major peaks connected to Ag presence are not visible in the diffractometer of Ag-olive.
The experiment part should be rewritten to apply all experimental details – preparation of the samples for SEM and TEM measurements, “the concentration were 60, 90 and 120 ppm” ppm of what? What does it mean “were isolated from Agriculture” – it should be specified.
Author Response
Reviewer #2: In my opinion, the manuscript needs to be extensively modified. There are some points that should be elucidated or improved and the study requires a major revision before publication. - Antiparasitic activity experiments were not described properly. 200 liters is a large amount of blood to test a single sample. “ 400 adult ticks were divided into four groups” What does it mean? - 4 samples of particles were immersed in blood and then … the ticks were put into the blood. – what does it mean “expose” in this experiment – it should be described in detail. The ticks need to sack blood from the host body. Probably the test found how many ticks survive in liquid and did not drow. The experiment should contain also the sample (control) ticks in blood without AgNPs. The key issue is also how antiparasitic properties of AgNPs will be applied, without in vivo experiments with host organisms is not possible to find proper concentrations of AgNPs, which do not harmfully influence animals. - First, we would like to thank the reviewer so much for the valuable comments. -Thank you for pointing it out and corrections. It was not 2 L, actually, 200 ul of blood were used. In response to your comments, the antiparasitic activity has been rectified. - The photocatalytic part should be also significantly improved. What was the wavelength of the lamp? The mechanism of photodegradation should be proposed. This part needs essential control experiments to prove the photodegradation process – according to the author's “solar degradation” - two different irradiation sources should be selected: from the visible part of spectra and from UV, the experiment without AgNPs to show that the dyes do not degradation in time or because of temperature? Moreover, an experiment carried out in the dark, without any lamp and irradiation is necessary to prove electromagnetic influence on the process. It is essential to prepare also experiment with commercial photocatalysts – P25 (TiO2) to compare the efficiency of the process. Why did methyl orange not decompose in the presence of AgNPs? -Thank you for pointing it out. Sunlight was only the source of light. For methyl orange dye, a special study will be conducted to study the effects of various factors such as pH, temperature, catalyst, and dye concentrations. In response to your comment, the photocatalytic part has been modified. - The authors should explain why the AgNPs size from Scherrer calculations is not comparable to SEM analysis. - Thank you for pointing it out. Whereas XRD measures the crystal size, SEM images detect the particle size. Particle size is larger than crystalline size. A grain or particle has a large number of crystals. So, the Ag-pure crystal's size was calculated to be 13.85 nm, and in SEM images, a diameter ranging from 48 to 95 nm was visible in the Ag-pure image. - Figures 5A and 5B need separate figure captions. - Thank you for pointing it out. In response to your comment, captions of Fig 5 have been updated. - Figure 2 – the main bands should be labeled in Figure and It should be explained why some major peaks connected to Ag presence are not visible in the diffractometer of Ag-olive. - Thank you for pointing it out. In response to your comment, the standard peaks of Ag-NPs have been illustrated in Fig.2. The main bands of Ag-NPs appear in the XRD of Ag-olive but are confused due to the presence of the peaks of amorphous organic phytocemicals. -The experiment part should be rewritten to apply all experimental details – preparation of the samples for SEM and TEM measurements, “the concentration were 60, 90 and 120 ppm” ppm of what? What does it mean “were isolated from Agriculture” – it should be specified. - Thank you for pointing it out. The concentrations were 60, 90, and 120 ppm, which refer to the concentrations of the tested samples (olive leaf extract, Ag-olive, and Ag-pure). In response to your comment, the experimental section on antimicrobial activity has been modified. Agriculture water refers to farm water.Round 2
Reviewer 1 Report
Comments and Suggestions for Authors
All the revisions have been checked.